# Zeb1 facilitates corneal epithelial wound healing by maintaining corneal epithelial cell viability and mobility

Yingnan Zhang[1,2,8], Khoi K. Do[1,8], Fuhua Wang[1,3], Xiaoqin Lu[1,4], John Y. Liu[1], Chi Li[1,4], Brian P. Ceresa[5], Lijun Zhang[6], Douglas C. Dean [1,4,7✉] & Yongqing Liu [1,4,7✉]

The cornea is the outmost ocular tissue and plays an important role in protecting the eye from environmental insults. Corneal epithelial wounding provokes pain and fear and contributes to the most ocular trauma emergency assessments worldwide. ZEB1 is an essential transcription factor in development; but its roles in adult tissues are not clear. We identify Zeb1 is an intrinsic factor that facilitates corneal epithelial wound healing. In this study, we demonstrate that monoallelic deletion of Zeb1 significantly expedites corneal cell death and inhibits corneal epithelial EMT-related cell migration upon an epithelial debridement. We provide evidence that Zeb1-regulation of corneal epithelial wound healing is through the repression of genes required for Tnfa-induced epithelial cell death and the induction of genes beneficial for epithelial cell migration. We suggest utilizing TNF-α antagonists would reduce TNF/TNFR1-induced cell death in the corneal epithelium and inflammation in the corneal stroma to help corneal wound healing.

[1] Department of Medicine, University of Louisville School of Medicine, Louisville, KY 40202, USA. [2] The Rosenberg School of Optometry, University of the Incarnate Word, San Antonio, TX 78229, USA. [3] Eye Institute and Eye Hospital of Shangdong First Medical University, 250021 Jinan, China. [4] James Brown Cancer Center, University of Louisville School of Medicine, Louisville, KY 40202, USA. [5] Department of Pharmacology and Toxicology, University of Louisville School of Medicine, Louisville, KY 40202, USA. [6] Department of Ophthalmology, Third People's Hospital of Dalian, Dalian Medical University, 116033 Dalian, China. [7] Department of Ophthalmology and Visual Sciences, University of Louisville School of Medicine, Louisville, KY 40202, USA. [8] These authors contributed equally: Yingnan Zhang, Khoi K. Do. ✉email: douglas.dean@louisville.edu; y0liu016@louisville.edu

The cornea is a transparent tissue of the eye that focuses light onto the retina and must remain essentially clear for an optimal vision to be achieved[1]. It is also the most sensitive tissue in the body due to the density of nerves[2]. To maintain transparency the cornea retains avascular and unscarred and preserves its highly organized anatomic structure[1]. The cornea is the outmost front element in the eye's optical system so that plays an important role in protecting the eye from environmental insults. To effectively treat corneal wounds to restore a clear vision, a comprehensive understanding of their etiology and the underlying mechanisms of wound healing are essential. The barrier function of the epithelium is directed against physical traumas, pathogens and chemicals to protect the underlying stroma that mainly contains quiescent keratocytes and it may lose its transparency if damaged due to the infiltration of immune cells and the activation of the keratocytes. In most healthy individuals, a sole corneal epithelial wound heals quickly and effectively usually within a couple of days and is expected to be fully recovered to the pre-injury status both structurally and functionally[3,4]. The major mechanism underlying this quick recovery of the damaged corneal epithelium is the existence of the limbal epithelial stem cells (LESCs) at the limbus[5]. Thus, unless the limbus is disrupted, the injury of the corneal epithelium is almost always limited and self-healed[1,4–6].

ZEB1 represses the epithelial gene E-cadherin (CDH1) and upregulates the mesenchymal genes vimentin (VIM) and N-cadherin (CDH2), thereby increasing epithelial cell mobility and proliferative ability[7]. It has been shown that Zeb1 is expressed at a lower level in the normal mouse corneal epithelial cells despite the LESC-derived epithelial basal cells keep dividing during the corneal epithelium homeostasis in contrast to the relatively quiescent stromal cells[3,8]. We therefore are wondering whether corneal epithelial cells would increase Zeb1 expression when they are stimulated by a wounding stress? And whether a higher level of Zeb1 expression in the epithelial cells is required for them to divide more frequently and migrate more quickly to recover the de-epithelialized area? Conversely, whether a reduction of Zeb1 would thereby delay the corneal wound healing? Here, we show that the mechanical debridement of the corneal epithelium, though decreases initially, increases Zeb1 expression thereafter in parallel with an augmentation of apoptotic and proliferative cells in the affected corneas. The monoallelic knockout (KO) of Zeb1 (Zeb1[+/−]) that results in partial loss of Zeb1 expression in mouse corneal epithelial cells reduces corneal epithelial cell viability and mobility, thereby delaying the corneal epithelial recovery after the de-epithelialization. We conclude that Zeb1 facilitates the corneal epithelial wound healing by repressing Tnfa-induced epithelial cell death genes and upregulating epithelial to mesenchymal transition (EMT), the extracellular matrix (ECM), ECM-degrading enzyme and cell-ECM anchor genes.

## Results

**Monoallelic Zeb1-KO reduces corneal re-epithelization.** Previously, we reported that Zeb1 promotes corneal inflammation after an alkali burn by promoting bone marrow-derived cell (BMC) viability and mobility[9], thereby facilitating corneal angiogenesis[10]. Angiogenesis has been reported to facilitate tissue wound healing[11,12]. However, whether Zeb1 facilitates the epithelial wound healing has not been tested. We therefore sought to investigate the epithelial wound healing after a mechanical removal of the epithelium in the central corneal area of 2-mm diameter (see the methods). To characterize corneal responses to the mechanical debridement, the denuded area of the epithelium was monitored by fluorescein staining and photographed under a fluorescent microscope over a period of 3 days. As a result, the

Zeb1[+/+] corneas healed quickly and the denuded area appeared almost completely recovered in 2 days after the debridement (Fig. 1a). To clarify whether Zeb1 regulates the corneal re-epithelization we checked how the heterozygous (het) Zeb1-knockout (KO) mice would respond to the de-epithelialization. As the embryos with the homozygous Zeb1-KO (Zeb1[−/−]) die before their birth, the heterozygous Zeb1-KO (Zeb1[+/−]) mice were utilized for the experiments herein. Compared to the wild-type (wt) Zeb1[+/+] corneas, the monoallelic deletion of Zeb1 significantly delayed the recovery of the debrided epithelium (Fig. 1b). Zeb1 has been reported to facilitate both cell proliferation and migration[9,10,13,14], thus Zeb1 may play a role in recovering the denuded corneal epithelium through promoting the proliferation and/or migration of the corneal epithelial cells[4].

**Mechanical debridement of the corneal epithelium induces cell death.** It has been reported that any physical trauma that would severely damage the cornea can result in corneal cell death[1]. Thus, we intended to check if the debridement would result in corneal cell death using the TUNEL method. As expected, the

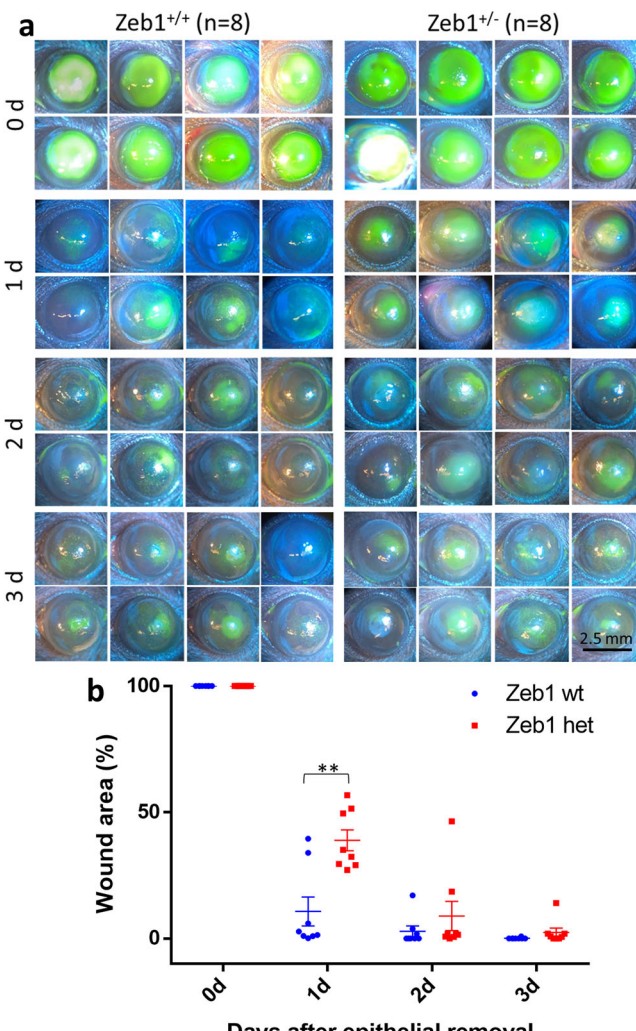

**Fig. 1 Monoallelic Zeb1-KO delays the recovery of the debrided epithelium. a** Images of the fluorescein-stained corneas of both Zeb1[+/+] and Zeb1[+/−] mice over period of 3 days after the epithelial debridement. **b** The debrided areas were measured with the software ImageJ and analyzed by the software GraphPad Prism. wt, Zeb1[+/+]; het, Zeb1[+/−]; **p ≤ 0.01; n, the number of corneas used. The bars are the mean ± standard error bars.

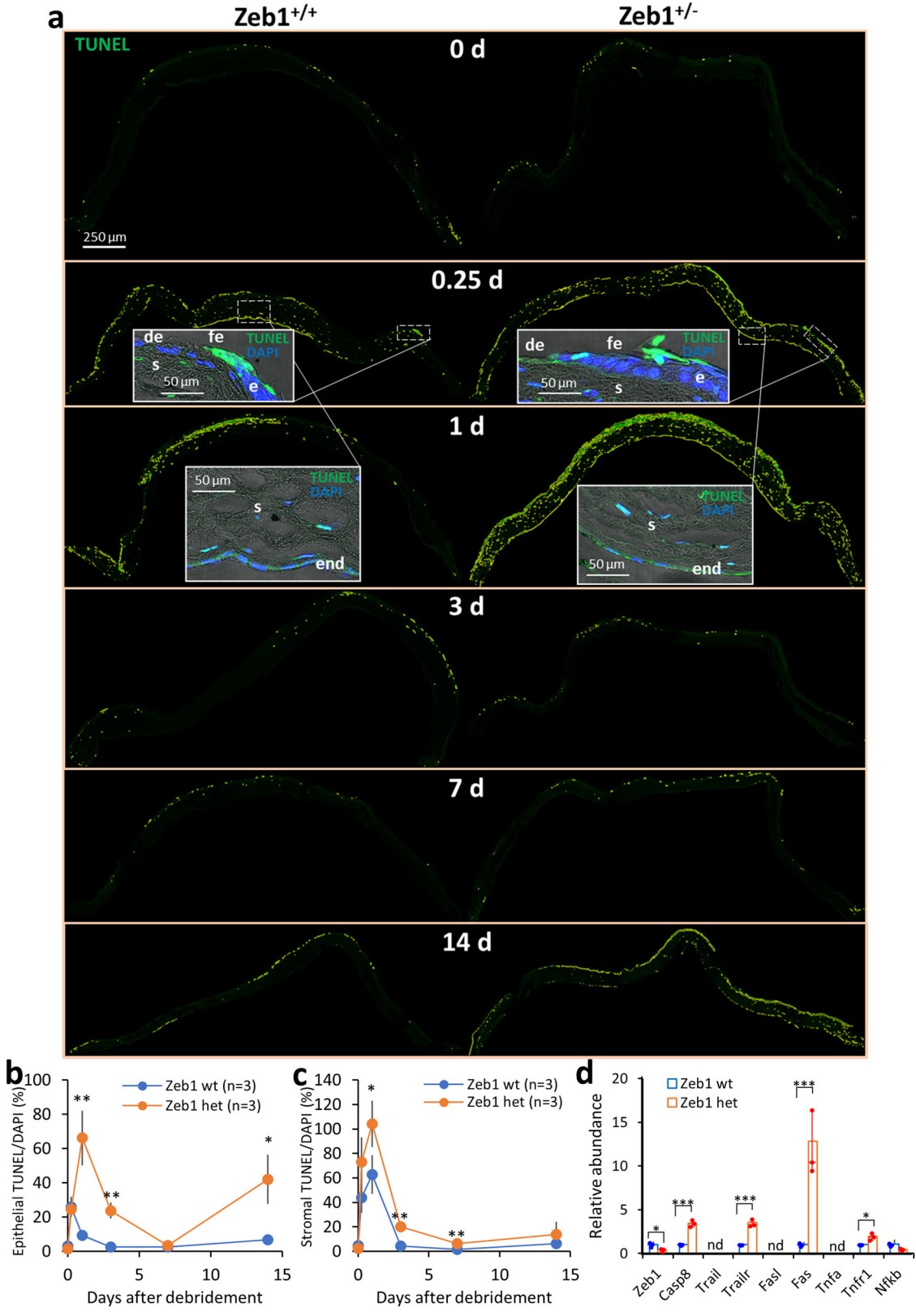

debridement induced an immediate death of large number of both epithelial and stromal cells in Zeb1+/+ corneas within one day after the debridement; the cell death rates declined thereafter (Fig. 2a–c). To validate this observation, we selectively conducted a wholemount immunostaining of both Zeb1+/+ and Zeb1+/− corneas on day 0 (before debridement) and day 1 after the

debridement to detect changes in corneal TUNEL+ cell death and had a result (Supplemental Fig. S1a–b) similar to the corneal sections (Fig. 2a–c). We next wanted to determine signals leading to the cell death. An increased cellular stress and disassociation from neighboring cells and the extracellular matrix (ECM) are possible causes[15]. The tumor necrosis factor (TNF) released from

**Fig. 2 Epithelial debridement induces corneal cell death and the monoallelic Zeb1-KO enhances the debridement-induced corneal cell death.**
**a** Representative ImageJ-processed images of TUNEL assays on the paraffin sections of Zeb1[+/+] and Zeb1[+/−] corneal tissues collected over period of 14 days after the epithelial debridement. Inserts, the detailed demonstration of the front edge of the debrided epithelium, the stroma and endothelium of both Zeb1[+/+] and Zeb1[+/−] corneas. Quantitative analyses on the cell death rates of the TUNEL+ cells to total DAPI+ cells and the comparison between the Zeb1[+/+] and Zeb1[+/−] corneal tissues in **b** the epithelium and **c** the stroma before (0 day) and after the debridement. **d** The expression of genes involved in the TNF/TNFR1 signaling pathway leading to the stress-induced cell death in the cultured primary epithelial cells of both wt and het corneas. wt, Zeb1[+/+]; het, Zeb1[+/−]; nd, not detected; de, de-epithelialized; fe, front edge of the debrided epithelium; e, the epithelium; s, the stroma; end, endothelium; *$p \leq 0.05$; **$p \leq 0.01$; ***$p \leq 0.001$; n, the number of corneas assessed. The error bars in **b** and **c** are the standard deviation (SD) bars whereas the box height in **d** is the mean with individual measurement points.

the broken basal membrane and/or secreted by the local and infiltrated immune cells and the corneal endothelial cells is one of the intrinsic factors causing the inflammation-associated apoptosis in the cornea[16]. Also, anoikis occurs when epithelial cells detach from their neighboring cells and the ECM[17]. Both may contribute to the debridement-induced corneal epithelial cell death, particularly those cells at the broken edge of the epithelium. As the death of the cells observed includes both the epithelial and the stromal cells, i.e., keratocytes and immune cells (Fig. 2c), we believe it is likely caused by the stress—the debridement and through the TNF/TNFR1-related signaling pathways[16]. Meanwhile, it is of note that a clear apoptotic spot on the broken edge of the epithelium in 6 h (0.25 day) after the debridement indicates that the anoikis may also exist as these epithelial cells are not surrounded by their neighboring cell and likely detached from the basal ECM (Fig. 2a, inserts).

**The epithelial debridement-induced Tnfa expression is likely the cause for the corneal cell death.** There are three major ligands for TNF/TNFR-related cell death pathways: TNF-α, TRAIL and FASL[18]. To determine whether TNF-α is present and whether the epithelial debridement would change the amount of the TNF-α protein in the cornea, we checked the dynamic expression of Tnfa in different parts of the mouse cornea after the debridement by immunohistochemistry of corneal cryosections (Fig. 3a). We found that the debridement immediately increased Tnfa expression in the cornea, including the epithelium, stroma and endothelium (Fig. 3a–d), implying Tnfa is likely of a soluble form[19] that diffuses into all parts of the cornea. To confirm the result, we also immunostained wholemount corneas with the Tnfa antibody and had a similar result (Supplemental Fig. S1a–b) to the corneal sections (Fig. 3a–c). Surprisingly, Tnfa concentrations increased from the epithelium to the stroma and to the endothelium (Fig. 3a–d, inserts), which was opposite to their order in cell death (Fig. 2a, inserts), indicating that the number of Tnf receptors on the epithelial cells are likely higher than that on both the stromal and endothelial cells, and Tnf is mostly produced by local and infiltrated immune cells in the stroma and by the endothelium as reported[20].

**Monoallelic Zeb1-KO increases corneal Tnfa expression and cell death rates.** To determine if Zeb1 regulates the epithelial debridement-induced corneal cell death, we checked the TUNEL-stained sections of both Zeb1[+/+] and Zeb1[+/−] corneas (Fig. 2a). Compared to Zeb1[+/+] corneas, the monoallelic Zeb1-KO significantly increased cell death rates in Zeb1[+/−] cornea, particularly in the epithelium (Fig. 2a–c), suggesting more epithelial cell death in Zeb1[+/−] corneas, contributing to the slower pace of the epithelial recovery after the debridement (Fig. 1). To determine whether Zeb1 regulates expression of Tnfa, we examined its presence in Zeb1[+/−] and Zeb1[+/+] corneal sections. The monoallelic Zeb1-KO significantly increased the amounts of the Tnfa protein in Zeb1[+/−] corneas compared to Zeb1[+/+] corneas (Fig. 3), implying an activation of the debridement-induced Tnfa/

Tnfr1 cell death signaling pathway[20]. To further verify this mechanism, we isolated corneal epithelial cells from both Zeb1[+/+] and Zeb1[+/−] mice (Supplemental Fig. S2a–d). These primary epithelial cells were of an epithelial morphology when they moved out of the explant corneas and adhered to culture plates (Supplemental Fig. S2e). We cultured them until confluence and verified their epithelial cell identity by staining with keratin 12 (Krt12) and E-cadherin (Cdh1) (Supplemental Fig. S2f–g). qPCR was performed to analyze their gene expression using total RNA isolated from both cultured Zeb1[+/−] and Zeb1[+/+] corneal epithelial cells. We found no detection of the ligands *Tnfa*, *Trail* and *Fasl* genes whereas the message RNA of their cognate receptors (e.g., *Tnfr1*, *Trailr* and *Fas*) was all high in the corneal epithelial cells (Fig. 2d). The *Fas* gene, which is the most effective Tnf receptor leading to cell death[18], was induced to the highest level (Fig. 2d). This is consistent with corneal epithelial cells not being the source of Tnfa (Fig. 3a, inserts), but likely the target of this paracrine factor. In addition, the monoallelic Zeb1-KO significantly increased the expression of all the TNF receptor genes *Tnfr1*, *Trailr* and *Fas* (Fig. 2d), suggesting an upregulation of these TNF receptors in the Zeb1[+/−] corneal epithelial cells is likely the cause for more cell death in the Zeb1[+/−] epithelium than that in the Zeb1[+/+] epithelium (Fig. 2a–b).

**The epithelial debridement increases corneal epithelial cell proliferation.** ZEB1 is an EMT factor and oncogene in tumorigenesis[7]. Its upregulation and activation are positively related to cell proliferation as it represses the expression of the epithelial gene *CDH1*, which inhibits cell proliferation and migration[7]. As a mitotic population, the corneal epithelial basal cells are regenerated by the limbal epithelial stem cells (LESCs), and thereby stained with the cell proliferative marker Ki67[5]. We noted that before the debridement (0 day), a few Ki67+ epithelial cells scattered along the epithelium of Zeb1[+/+] corneas (Fig. 4a, inserts), indicating a normal cell homeostasis. The number of Ki67+ cells were immediately increased in Zeb1[+/+] corneas after the debridement and more Ki67 cells was detected close to the limbus (Fig. 4b, c) when Zeb1 expression in the cornea was actually downregulated at the same time (Fig. 4d). No positive correlation between the levels of Zeb1 expression and the number of Ki67+ cells was found in the cornea after the epithelial debridement (Fig. 4c, d). Also, no significant difference in the number of Ki67+ cells was found between Zeb1[+/+] and Zeb1[+/−] corneas throughout the first 3 days after the debridement (Fig. 4c and Supplemental Fig. S3). However, the proliferation rates of the epithelial cells in the Zeb1[+/+] corneas was significantly increased on day 7 and thereafter whereas no such increase was detected in the Zeb1[+/−] corneas (Fig. 4c), suggesting that the monoallelic Zeb1-KO reduces the proliferation of the corneal epithelial cells only in the later phase of the wound healing, and thereby should not negatively affect the re-epithelialization in the early phase of the wound healing (Fig. 1). This speculation was supported by the observation that Zeb1 expression in the Zeb1[+/+] and Zeb1[+/−]

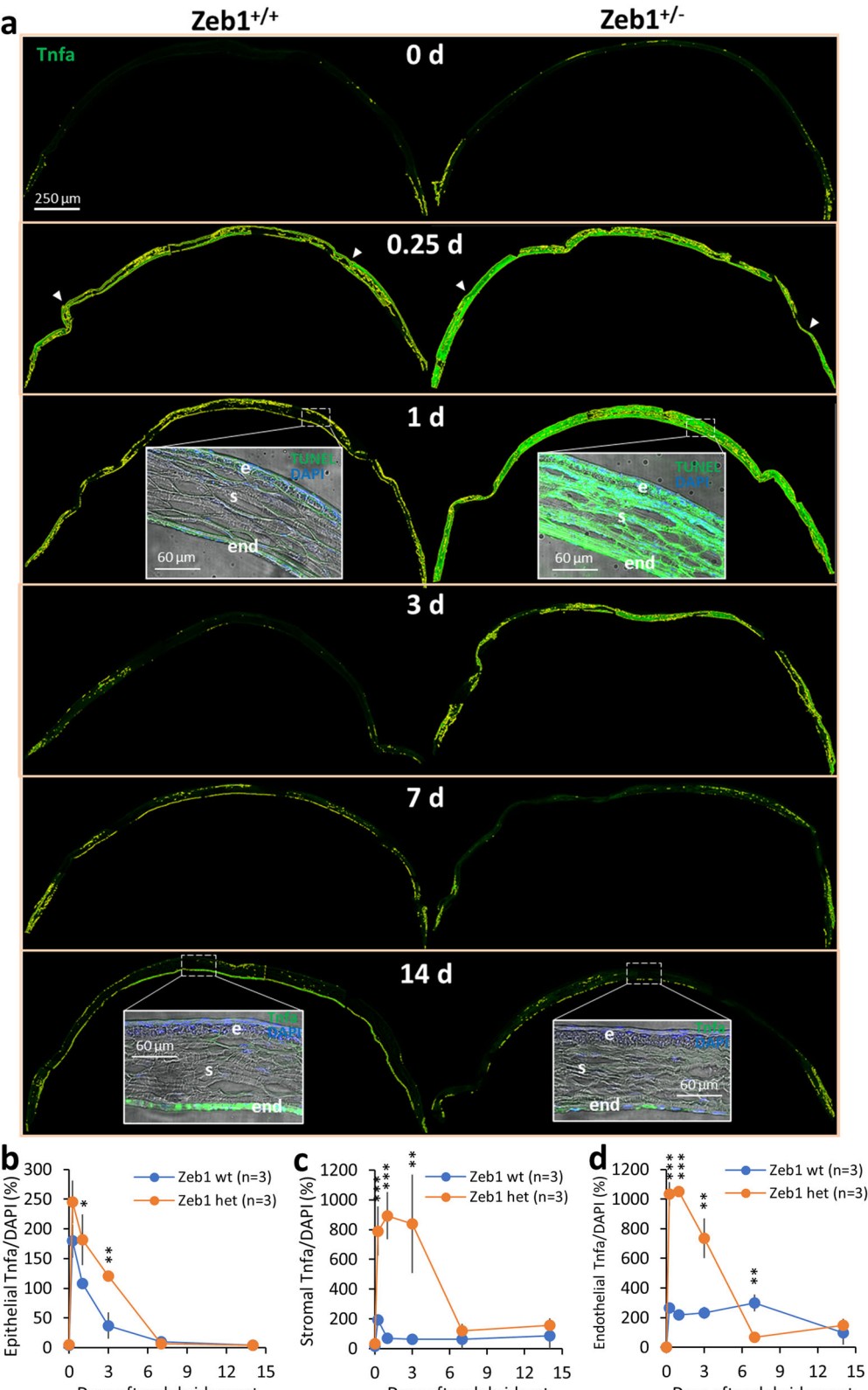

**Fig. 3 Epithelial debridement increases Tnfa expression and the monoallelic Zeb1-KO enhances the debridement-induced Tnfa expression in the cornea. a** Representative ImageJ-processed images of the Tnfa expression on the cryosections of $Zeb1^{+/+}$ and $Zeb1^{+/-}$ corneal tissues collected over period of 14 days after the epithelial debridement. The 14-day $Zeb1^{+/-}$ cornea may not be representative as it was the only complete cornea while the other two corneas were broken and incomplete. The quantitative analyses on the Tnfa expression to the total DAPI + staining and the comparison between the $Zeb1^{+/+}$ and $Zeb1^{+/-}$ corneal tissues in **b** the epithelium, **c** the stroma, and **d** the endothelium. Inserts, the detailed expression of Tnfa in different corneal cellular layers after the epithelial debridement; wt, $Zeb1^{+/+}$; het, $Zeb1^{+/-}$; inverted white triangles, wound borders; e, the epithelium; s, the stroma; end, the endothelium. $*p \leq 0.05$; $**p \leq 0.01$; $***p \leq 0.001$; $n$, the number of corneas used. The error bars in **b**, **c**, and **d** are the standard deviation (SD) bars.

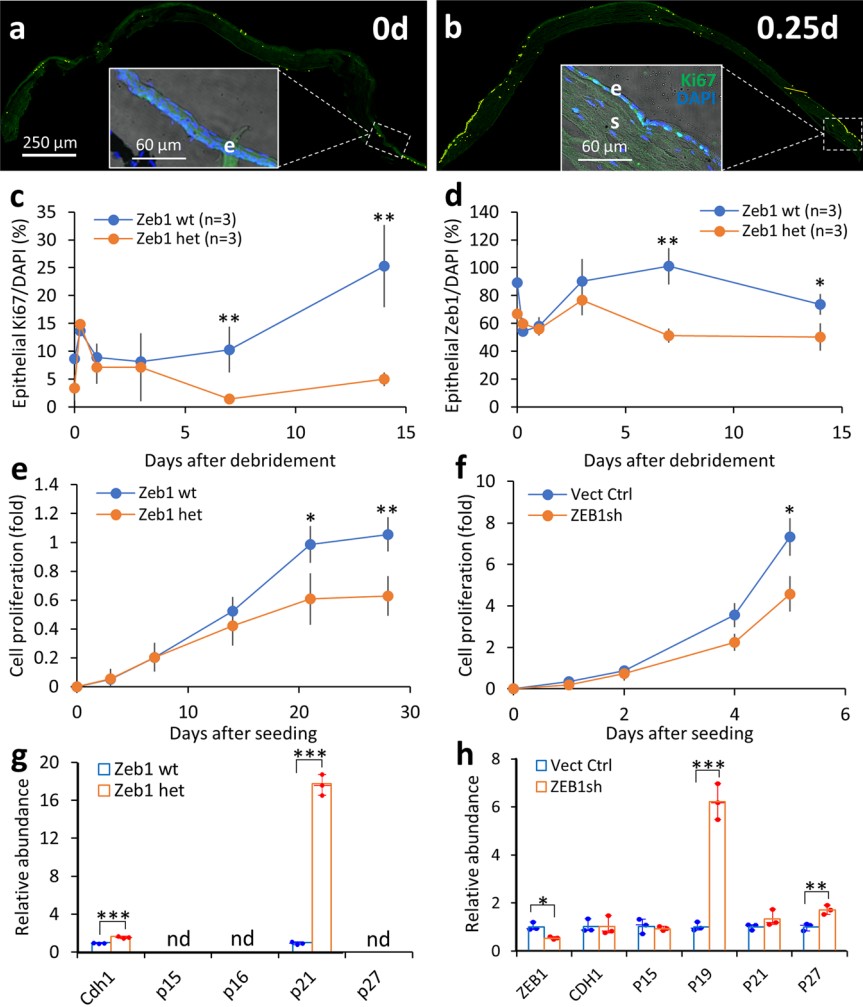

**Fig. 4 KO or knockdown (KD) of ZEB1 reduces corneal epithelial cell proliferation.** A representative image of the Zeb1$^{+/+}$ corneal tissue sections stained with the cell proliferative marker Ki67 **a** before (0 day) and **b** on day 1 after the epithelial debridement. Inserts, the epithelium close to the limbus. For the period of 14 days after the debridement, the ratios of **c** the Ki67+ or **d** the Zeb1+ area to the DAPI+ area in the epithelium of the Zeb1 wt corneas were compared with that of the Zeb1 het corneas. The comparisons of proliferation rates **e** between Zeb1 wt and Zeb1 het mouse primary corneal epithelial cells and **f** between vector control (Vect Ctrl) and ZEB1-KD (ZEB1sh) human hTCEpi cells. Relative expression levels of genes involved in cell proliferation were compared **g** between Zeb1 wt and Zeb1 het mouse corneal epithelial cells and **h** between Vect Ctrl and ZEB1sh hTCEpi cells. wt, Zeb1$^{+/+}$; het, Zeb1$^{+/-}$; nd, not detected; e, the epithelium; s, the stroma; *$p \leq 0.05$; **$p \leq 0.01$; ***$p \leq 0.001$; $n$, the number of corneas used. The error bars in **c**–**f** are the standard deviation (SD) bars whereas the box height in **g** and **h** is the mean with individual measurement points.

corneal epithelium detected by the immunostaining was actually decreased right after the debridement (Fig. 4d and Supplemental Fig. S4) in opposition to the increase in the epithelial cell proliferation rates (Fig. 4c). It appears that the remaining epithelial cells right after the debridement are being signaled to be apoptotic when new TA cells have not been regenerated yet by LESCs. Considering less Ki67+ epithelial cells in the Zeb1$^{+/-}$ epithelium compared to the Zeb1$^{+/+}$ epithelium (Fig. 4c), we speculate that cellular homeostasis in the Zeb1$^{+/-}$ epithelium is slower than that in the Zeb1$^{+/+}$ epithelium. So far, it is not clear whether this slower cellular homeostasis would affect corneal normality or not.

**Zeb1-KO and ZEB1-knockdown (KD) reduces corneal epithelial cell proliferation rates in vitro.** To clarify whether Zeb1 positively or negatively affect corneal epithelial cell proliferation, we cultured mouse epithelial cells isolated from both Zeb1$^{+/+}$ and Zeb1$^{+/-}$ corneas (Supplemental Fig. S2) to check their proliferation rates. The primary mouse corneal epithelial cells grew slowly though Zeb1$^{+/+}$ cells divided more frequently than Zeb1$^{+/-}$ cells (Fig. 4e). To validate this result, we also infected the

human telomerase-immortalized corneal epithelial cell line (hTCEpi) cells with the ZEB1 short hairpin interfering RNA (shRNA) lentivirus to knockdown ZEB1 in the cells (Supplemental Fig. S5). The infection rates were about 80% based on the GFP expression of the cells for both vector control (Vect Ctrl) and ZEB1-KD cells (ZEB1sh) before the first passage (P0) (Supplemental Fig. S5). ZEB1-KD was confirmed by immunofluorescence assays (Supplemental Fig. S6) and qPCR (Fig. 4h). The intensities of the ZEB1 and Ki67 immunostaining in hTCEpi cells with the ZEB1sh GFP was significantly lower than the rest cells without the ZEB1sh GFP or compared to the cells with the vector control GFP (Supplemental Fig. S6). The proliferation rate of the ZEB1sh cells was significantly declined compared to the vector control cells (Fig. 4f). Taken together, we conclude that ZEB1, as expected, promotes the proliferation of the corneal epithelial cells, no matter if it is a mouse or human origin. The reason why ZEB1 can promote cell proliferation often may as we previously reported[10,13,14], be its repression of cyclin-dependent kinase (CDK) inhibitors. Thus, we checked these CDK inhibitors using qPCR[9,10,14], and found the expression of *p21*, in addition to

the E-cadherin gene *Cdh1* in the mouse primary corneal epithelial cells (Fig. 4g) and the expression of *P19, P21* and *P27*, in addition to *CDH1* in the human hTCEpi cells (Fig. 4h), were significantly upregulated by the reduction of ZEB1 (Fig. 4g), confirming ZEB1 promotes corneal epithelial cell proliferation by repressing cycling inhibitor genes[10,13,14].

**The epithelial debridement reduces the corneal epithelial cell-cell adherent Cdh1.** As reported above, Zeb1 promotes corneal re-epithelialization independent of cell mitosis. We therefore postulate that Zeb1 facilitates corneal epithelial wound healing by promoting cell migration in addition to reducing cell death. It has been reported that corneal re-epithelization after a debridement is largely achieved by the extension of the remaining epithelial cells to recover the denuded area[21,22]. There are two models for epithelial cells to migrate: (1) the movement of the continuous sheets of epithelial cells[23,24] and (2) the transition of epithelial cells to mesenchymal cells (EMT), thereby facilitating their individual cell mobility[1,24]. Individual corneal epithelial cells are bound together by the adhesion protein E-cadherin (CDH1), thereby typically immobile[1,25,26]. For the corneal epithelial cells to become mobile individually, cell junction proteins like CDH1 must be degraded and transit to reorganize their cytoskeleton to extend the lamellipodia-like protrusions supported by cytoskeletal proteins like vimentin (VIM)[27]. In addition, to recover the epithelium the epithelial cells have to produce more ECM-degrading enzymes like the plasminogen activator urokinase (PLAU) and matrix metalloproteinases (MMP) to release the epithelial basal cells from the ECM, more ECM proteins like fibronectin (FN) to lay down a suitable foundation for new epithelium reestablishment, and more integrins (ITG) to adhesively interact with the substratum[1,27–29]. However, for the movement of the continuous sheet, corneal epithelial cells do not need change their cell-cell junctions. Or otherwise, they must adopt EMT to facilitate their migration from the front edge of the debrided epithelium to recover the denuded area. To determine how E-cadherin and Vimentin, the well-known cell-to-cell connection components regulated by Zeb1, are changed in expression over the recovery of the denuded epithelium, we immunostained the corneal cryo-sections with the epithelial marker Cdh1 and the mesenchymal marker Vim (Fig. 5a, b). We found before the epithelial debridement (0 day), Cdh1 was highly present in the basal cells along the epithelium while no Vim was detected (Fig. 5a, b). In 6 h (0.25 day) after the debridement, the epithelial cells along the front edge of the debrided epithelium were stained with Vim (Fig. 5b), whereas those behind the debrided edge were still Cdh1+ (Fig. 5a), suggesting an apparent EMT occurring only along the front edge of the epithelial debridement (Fig. 5a). However, this EMT was transient and immediately diminished when the moving epithelial basal cells recovered the denuded area in 1 day after the debridement (Fig. 5b). This transient EMT was likely due to a transient decrease in Cdh1 in the epithelium right after the debridement (Fig. 5c).

**Monoallelic Zeb1-KO augments Cdh1 in the corneal epithelium.** As ZEB1 is a well-known *CDH1* repressor, the monoallelic Zeb1-KO in the Zeb1$^{+/-}$ corneas significantly increased the amounts of Cdh1 in both the epithelium and the endothelium (Fig. 5c, d). Accordingly, no Vim and thereby EMT, was detected in the Zeb1$^{+/-}$ epithelium (Fig. 5b). Thus, the movement of the Zeb1$^{+/-}$ corneal epithelial cells on the ECM is likely not dependent on the EMT-related individual cell mobility; but relies on the continuous sheet sliding and thereby it is slower than the Zeb1$^{+/+}$ corneal epithelial cells with the capacities of both continuous cell sheet movement and individual cell migration. The

debridement could not reduce enough epithelial Cdh1 in Zeb1-KO mice to the threshold for the EMT to occur in the epithelium, thereby resulting in retardment of the epithelial cell movement. This is likely one of the major mechanisms for Zeb1 to positively regulate the corneal epithelial recovery after the debridement (Fig. 1).

**ZEB1 promotes corneal epithelial cell migration.** To mimic the movement of the corneal epithelial cells in vitro, we monolayer-cultured Zeb1$^{+/+}$ and Zeb1$^{+/-}$ mouse primary corneal epithelial cells and checked their migration rates by a scratch assay as previously reported[9,10]. We found the monoallelic Zeb1-KO significantly reduced cell migration rate in culture (Fig. 6a, b). To verify the result, we also checked the migration rates of the vector control and ZEB1-KD hTCEpi cells and found ZEB1-KD significantly reduced cell migration (Fig. 6c, d). These two experiments suggest that ZEB1 is an important factor in promotion of corneal epithelial cell migration, confirming above in vivo observations on the Zeb1-regulated corneal epithelial EMT (Fig. 5).

**Monoallelic Zeb1-KO or ZEB1-KD reduces the expression of genes involved in corneal epithelial cell migration.** Independent of whether corneal epithelial cells move as continuous sheets or by lamellipodia-like protrusion, the cell movement depends on sliding on the ECM protein-rich substratum[23]. Integrin hetero-dimers on epithelial cell surface are the major anchor for the ligands that are present on substratum proteins like fibronectin (FN)[30]. To validate the mechanism underlying the promoting effects of Zeb1 on mouse epithelial cell EMT and migration in vivo, we checked the expression of the related genes in the cultured Zeb1$^{+/-}$ corneal epithelial cells and ZEB1-KD hTCEpi cells compared to their wildtype or vector control, respectively. In both mouse and human corneal epithelial cells, the reduction of ZEB1 increased *CDH1* and decreased *VIM* as expected (Fig. 6e, f), suggesting an obvious EMT switch under ZEB1 control. The reduction of ZEB1 also decreased the expression of the ECM gene *FN1* and the ECM modifying genes *PLAU* and *MMP* (Figs. 4g, h and 6e, f). Although the expression of integrin (ITG) genes including alpha and beta subunits was not detected in the mouse primary corneal epithelial cells, integrin alpha 5 (*ITGA5*) was highly expressed in hTCEpi cells, and ZEB1-KD reduced its expression (Fig. 6f). Taken together, ZEB1 is an important factor in regulation of corneal epithelial cell movement on the substratum likely by switching on the transient EMT, degrading cell-substratum connection, renewing the ECM and re-establishing new cell-ECM connections.

**Molecular mechanisms underlying Zeb1 regulation of corneal epithelial wound healing.** We have shown above that Zeb1 promotes mouse corneal epithelial wound healing by reducing epithelial cell death through the repression of TNF/TNFR1 signaling pathway (Fig. 2)[16]. To clarify whether Zeb1 directly binds to and thereby regulates the expression of *Tnfa* and/or *Tnfr1* gene, we identified multiple sites of the Zeb1 DNA binding consensus sequence (CANNTG) including a perfect match sequence (CACCTG) in each of their putative promoter regions. However, we did not detect any binding of Zeb1 to the promoter of either *Tnfa* or *Tnfr1* by the chromatin immuno-precipitation (ChIP) assay (Fig. 7a)[31], implying Zeb1 indirect regulation of Tnf/Tnfr1 signaling pathway. TNF-α is also a pro-inflammation cytokine to activate the inflammation master regulator NFκB leading to tissue inflammation and cell survival[32]. This is a two-way valve: one way for cell death with the help of caspase 8 (CASP8) whereas the other is for inflammation and cell

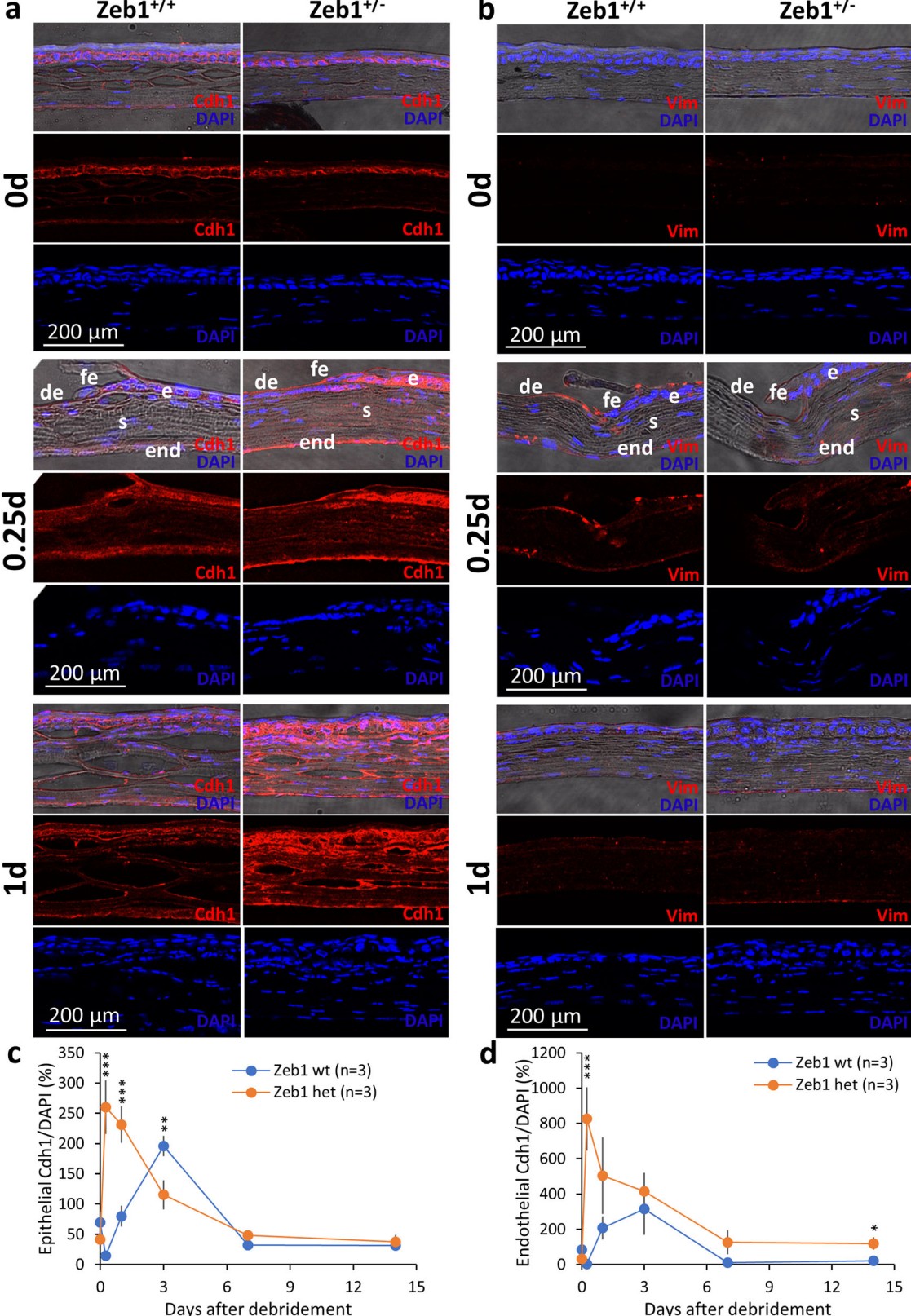

**Fig. 5 Monoallelic Zeb1-KO inhibits corneal epithelial EMT.** The representative images of **a** Cdh- and **b** Vim-stained cryosections of both $Zeb1^{+/+}$ and $Zeb1^{+/-}$ corneas before (0 day) and after the epithelial debridement. The amounts of Cdh1 per DAPI unit detected by immunohistochemistry and measured by ImageJ in **c** the epithelium and **d** the endothelium before (0 day) and after the epithelial debridement. wt, $Zeb1^{+/+}$; het, $Zeb1^{+/-}$; de, de-epithelialized; fe, front edge of the debrided epithelium; e, the epithelium; s, the stroma; end, the endothelium; *$p \leq 0.05$; **$p \leq 0.01$; ***$p \leq 0.001$. The error bars are the standard deviation (SD) bars.

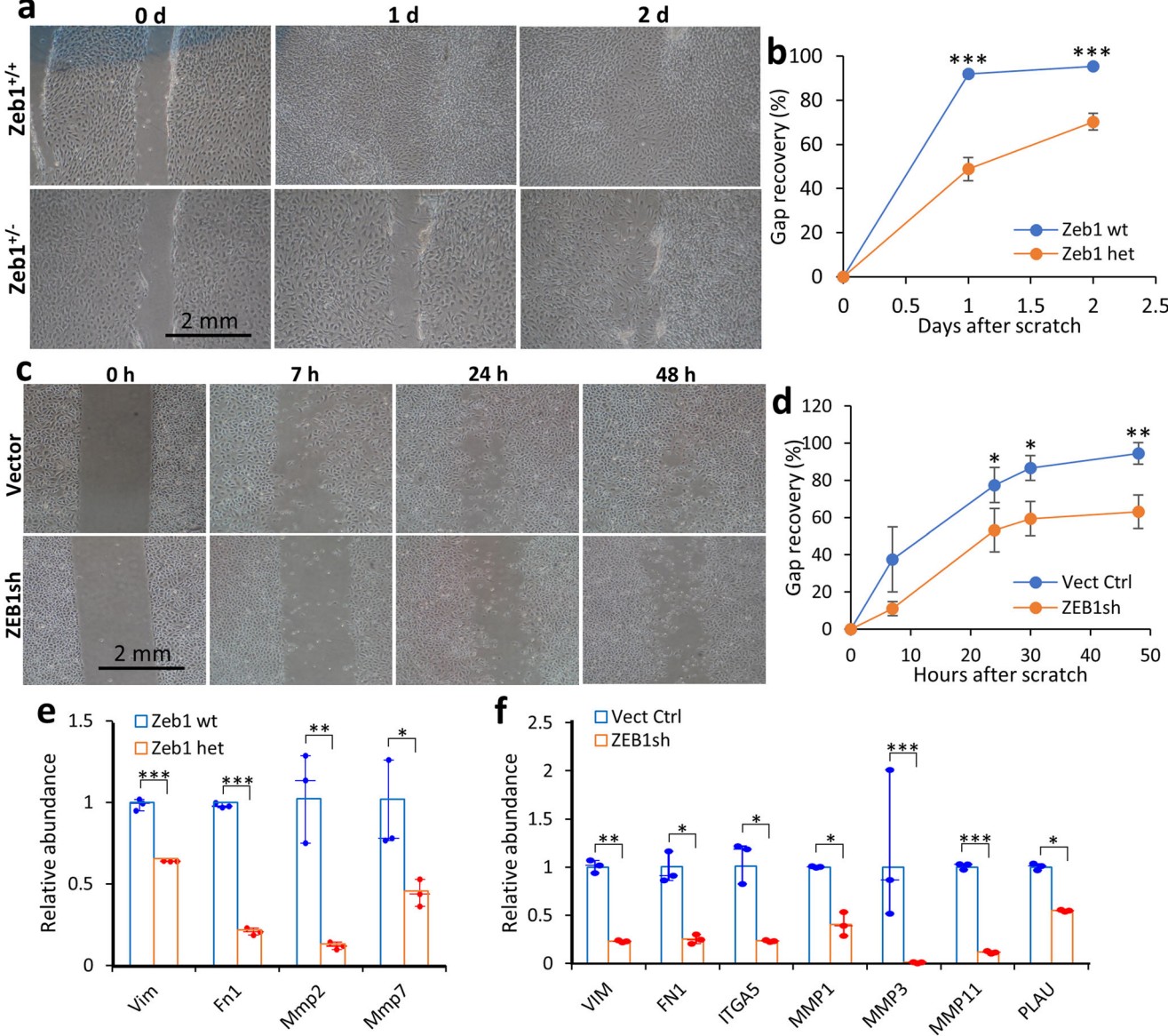

**Fig. 6 Zeb1 promotes corneal epithelial cell migration. a** The representative images of the epithelial cell monolayer cultures of both Zeb1$^{+/+}$ and Zeb1$^{+/-}$ corneas and **b** their migration rates. **c** The representative images of the hTECpi cell monolayer cultures of both the vector control (Vect Ctrl) and Zeb1-KD (ZEB1sh) and **d** their migration rates. **e** The monoallelic Zeb1-KO (Zeb1 het) in the mouse primary corneal epithelial cells and **f** ZEB1-KD (ZEB1sh) in the hTCEpi cells reduced the expression of cell migration-related ECM, the cell-matrix anchor and the matrix degradation enzyme genes in both mouse and human corneal epithelial cells. wt, Zeb1$^{+/+}$; het, Zeb1$^{+/-}$; *$p \leq 0.05$; **$p \leq 0.01$; ***$p \leq 0.001$. The error bars in **b** and **d** are the standard deviation (SD) bars whereas the box height in **e** and **f** is the mean with individual measurement points.

survival with the help of NFκB[18]. We therefore checked whether Zeb1 alternatively upregulates *Nfκb* to implement inflammation and cell survival functions. We found that the monoallelic Zeb1-KO reduced *Nfκb*, but upregulated *Casp8* (Fig. 2d), suggesting this Tnf/Tnfr1 two-way valve favors cell death in the mouse corneal epithelial cells. In addition, Zeb1 was also detected binding to the putative *Nfκb* promoter region (Fig. 7a), likely to transactivate *Nfκb* and thereby the Nfκb-regulated inflammation induction[9].

Further, it has also been shown that Zeb1 binds to and represses *Cdh1*[33], *Cdk* inhibitors (*p15, p19, p21, p27*)[10,13,14] to promote cell proliferation. We found that the monoallelic Zeb1-KO upregulated the expression of *Cdh1* and *p21* in the mouse primary corneal epithelial cells (Fig. 4g), confirming Zeb1 promotes corneal epithelial cell proliferation through binding and thereby repressing *CDH1* and *P21* (Fig. 7b). Downregulation

of *CDH1* while transactivating the mesenchymal cell marker *VIM* by ZEB1 is a prior event for EMT[33], which facilitates the recovery of the injured corneal epithelium (Fig. 7c). To be mobile, in addition to EMT, the corneal epithelial cells need to secrete more ECM-degrading enzymes like MMP and PLAU to release epithelial cell bonds to the substratum, more ECM proteins like FN1 to replace the degraded ECM and more cell-substratum anchors like ITG to re-establish their bonds[9,34], thereby facilitating the epithelial wound healing (Fig. 7c). Indeed, we have shown above Zeb1 upregulates *Mmp, Fn1* and *ITGA5* (Fig. 6e, f). To clarify whether these genes are directly bound and thereby regulated by Zeb1, we performed ChIP assays. As a result, we detected that ZEB1 binds to the putative promoters of the ECM-degrading enzyme *MMP11* and *PLAU* and the ECM *FN1* genes (Fig. 7b)[13], suggesting ZEB1 binds and likely transactivates these genes to promote corneal epithelial wound healing.

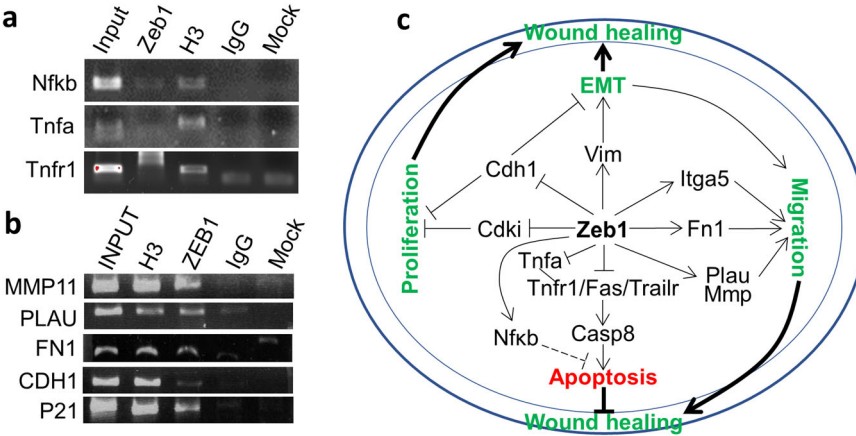

**Fig. 7 ZEB1 regulates the expression of genes involving corneal epithelial wound healing. a** Chromatin immunoprecipitation (ChIP) assays on the binding of mouse Zeb1 to the putative promoters of *Nfkb, Tnfa*, and *Tnfr1*. **b** ChIP assays on the binding of human ZEB1 to the putative promoters of *MMP1, PLAU, FN1, CDH1*, and *P21*. **c** A schematic diagram describing the molecular mechanisms underlying Zeb1-regulation of corneal epithelial wound healing.

## Discussion

ZEB1 is a transcription factor that regulates expression of many genes involving in cell proliferation, differentiation, mobility and viability[7]. ZEB1 is essential in the embryonic development as homozygous Zeb1-KO would result in embryonic death before birth[35]. In adult tissues, ZEB1 is either not expressed or expressed at low levels. Immunostaining of the corneal paraffin sections shows that Zeb1 protein presents in the corneal epithelium, in both the cytoplasm and the nucleus of the epithelial cells (Supplemental Fig. S4). Although the monoallelic Zeb1-KO delayed the epithelial wound healing after the debridement (Fig. 1), the reduction of Zeb1 seemingly did not affect the epithelial proliferation rates right after the debridement (Fig. 4c). Therefore, the Zeb1-regulation of corneal epithelial cell proliferation does not account for the quicker epithelial wound healing pace. Then, how does the monoallelic Zeb1-KO delay the recovery of the debrided epithelium? We demonstrate that Zeb1 promoted corneal epithelial wound healing likely through enhancing epithelial cell mobility (Fig. 6) in addition to maintaining epithelial cell viability (Fig. 2). Although the amounts of Zeb1 in the epithelium seemingly did not affect the epithelial cell proliferation in the early epithelial recovery after the debridement it did correlate with the proliferation of the epithelial cells in the later phase of the wound healing (Fig. 4c, d). In addition, the reduction of Zeb1 results in more epithelial cell death and less regeneration, thereby may negatively affect corneal epithelial homeostasis. Also, the monoallelic Zeb1-KO diminished the EMT along the front edge of the debrided epithelium and enhanced the connection between epithelial cells (Fig. 5a, c), and thereby may slow the sliding pace of the continuous epithelial sheet. We provided evidence that the monoallelic Zeb1-KO in mouse primary epithelial cells and ZEB1-KD in the human hTCEpi cells significantly reduced the expression of genes involving cell-cell adhesions, ECM proteins, ECM-degrading enzymes and cell-ECM anchors (Fig. 6e, f), and thereby resulting in the retardment of epithelial cell migration.

Zeb1 presents in the corneal stroma at an undetectable level (Supplemental Fig. S4). However, the epithelial debridement immediately stimulates the augmentation of Zeb1 in the stroma where both the infiltrated immune cells and local keratocytes are present in contrast to the reduced expression of Zeb1 in the epithelium (Supplemental Fig. S4). We recently reported that Zeb1 promotes corneal inflammation after an alkali burn by accelerating circulation of immune cells from the bone marrow to the injured cornea[9]. These infiltrated leukocytes could secrete large amounts of pro-inflammatory cytokines in the cornea[9] to

recruit more leukocytes and eventually leads to neovascularization in the affected cornea[15]—a serious pathogenic condition that reduces the visual acuity and even blindness. In this study, we also found the debridement-induced Zeb1 augmentation in the stroma (Supplemental Fig. S4) was positively correlated with the increase in the inflammation master regulator Nfkb, thereby supporting our previous finding that Zeb1 promotes corneal inflammation[9]. Based on the expression of Zeb1 (Fig. 4d and Supplemental Fig. S4), we classify the epithelial debridement-induced wound healing into two distinct phases. In the early phase, Zeb1 is downregulated in the epithelium and upregulated in the stroma when cell death and proliferation are induced in the epithelium and inflammation in the stroma. In the late phase, the expression of Zeb1 is up in the epithelium to the prior level when corneal cell death and proliferation are gradually declined.

ZEB1 is involved in cell differentiation and transformation in development and pathogenesis of many diseases including cancer and tissue fibrosis[7]. Mutations of ZEB1 have been linked to stem cell inefficiency[36], immune deficiency[37] and corneal endothelial dystrophy[38,39]. In adult tissues, abnormal activation of ZEB1 often results in cancer metastasis[40] and scar formation[41] that lead to tissue malfunction. ZEB1 has been reported to regulate expression of inflammatory cytokines like IL-6 and IL-8[42]. Normal inflammation is required to kill and remove pathogens and transformed cells from healthy tissues and to assist tissue wound healing[43]. Prolonged and/or over-run inflammation may damage the affected tissue, leading to the malfunction of related organs[44]. It is critical to modulate tissue inflammation such as by corticosteroids to avoid tissue damage[45]. However, use of steroids in ocular tissues may cause serious adverse side effects including glaucoma and cataracts[45]. Non-steroidal anti-inflammatory drugs are therefore the alternative solutions[45]. We recently have identified that ZEB1 is a major driver for promoting corneal inflammation and neovascularization (NV), and inhibition of ZEB1 by the ZEB1-CtBP inhibitor NSC95397 significantly reduces corneal NV, thereby rescues the affected visual acuity[10]. Zeb1 promotes corneal epithelial wound healing as we report herein, the application of NSC95397 to treatment of corneal NV may therefore be complicated with compromising corneal wound healing. However, we found the reduction of the epithelial cell viability by the monoallelic Zeb1-KO was the mechanism underlying the Zeb1-regulation of corneal epithelial wound healing through the TNF/TNFR1 apoptosis signaling pathway (Figs. 2 and 3). To reduce the debridement-induced corneal epithelial cell death and thereby expediting corneal epithelial

wound healing, specifically neutralizing Tnfa/Tnfr1 should be tested by directly applying the according antibodies or antagonists to the cornea after the mechanical epithelial damage in future medical applications.

## Methods

**Mouse model of the corneal epithelial wound healing**. Eight 3-month-old Zeb1 wild-type (wt, Zeb1$^{+/+}$) or heterozygous mutant (het, Zeb1$^{+/-}$) mice of mixed sexes with the background of C57BL/6J are anesthetized by an intraperitoneal (IP) injection of 100 mg/kg ketamine and 5 mg/kg xylazine. A circular area of 2 mm in diameter of the central corneal epithelium was marked using a trephine and debrided using the Alger Brush II under a stereo microscope. Only one cornea was debrided whereas the other eye remained untreated according to the Association for Research in Vision and Ophthalmology (ARVO)'s regulation. The epithelial wound of the same mice was evaluated by photography in a blinded manner using the fluorescein sodium ophthalmic strip in 0, 1, 2, and 3 days after the debridement. The mechanical debridement was confirmed by the H&E histology at 6 h (Supplemental Fig. S7) after the debridement. The fluorescein intensity of wound area was measured using the software ImageJ after cropping the entire corneal area and setting a fixed threshold and analyzed by the software GraphPad Prism. This animal study was conducted according to the policies and guidelines set forth by the Institutional Animal Care and Use Committee (IACUC) and approved by the University of Louisville, Kentucky, USA.

**Mouse corneal epithelial cell isolation and culture**. The eyeballs of 3–4-week-old pups were enucleated and thoroughly rinsed with sterile PBS. Corneal epithelial cells from younger animals could be passaged more times than older animals. The entire dome-like cornea was cut off and cut in quarters of a butterfly-like shape using a pair of scissors and then placed upside down on a culture dish with just enough culture medium (Cell Biologics, Cat. # M6621-Kit) to be uncovered and touched to the dish bottom. The epithelial cells migrated out of 6 corneas isolated from either Zeb1$^{+/-}$ or Zeb1$^{+/+}$ mice in a 6-cm dish and became a monolayer and confluent in 3–4 weeks and ready for cell migration assay and total RNA extraction.

**Knockdown (KD) of ZEB1**. The lentiviral constructs with the short hairpin interfering RNA (shRNA) sequence (5'-AACAATACAAGAGGTTAAA-3') against human *ZEB1* gene and a scramble fragment against nothing were purchased from Shanghai GenePharma Co Ltd (Shanghai, China). The lentivirus particles of these *ZEB1*-shRNAs and scramble shRNA together with an GFP gene were assembled in the laboratory as described previously[13]. ZEB1-KD by the construct was validated in the human uveal melanoma cell line C918 by a Western blot (WB) previously[13]. The human telomerase-immortalized corneal epithelial (hTCEpi) cells, a gift from Dr. Brian Ceresa at the University of Louisville[46], were cultured in the epithelial cell culture medium (Cell Biologics, Cat. # 220-500 plus # 221-GS) and infected separately with the *ZEB1*-shRNA and the scrambled GFP vector lentivirus. Based on the expression of GFP under a fluorescent microscope, the transduction rates of both the *ZEB1*-shRNA and the scrambled GFP vector were all above 80% at passage 0 (P0). However, we used the cells of passage 3 (P3) for the experiments when the percentage of the GFP cells was declined to about 60%. The KD of ZEB1 in the hTCEpi cells was validated in the transduced cells by qPCR.

**Cell migration assay**. Monolayer-cultured mouse primary corneal epithelial and hTCEpi cells at 100% confluence were treated with 5 μg/ml mitomycin C for 2 h at 37 °C and then washed with PBS, followed by a straight scratch using a 100 P or 1000 P pipette tip for the mouse or human epithelial cell monolayers, respectively, and photographed under an inverted microscope on the desired days at the same location. The width of 4 scratched gaps for each treatment was measured using ImageJ and the invert of the measurement was served as a gap closing rate.

**Immunohistochemistry (IHC)**. The eyeballs were enucleated and fixed in 10% formalin for paraffin-section, or in 4% paraformaldehyde for cryosection as previously reported[10]. All sections are sagittal cuts to ensure that the anterior central cornea and the posterior optical nerve bundle are in the same plane. For corneal wholemount TUNEL detection (see below) and Tnfa immunostaining, after a brief rinse with PBS and removal of the retina, lens and iris, the circular corneas with the limbus were dissected from the eyeballs and transferred to a 96-well plate. The stained corneas were thereafter placed on a glass slide with the epithelium layer facing up and cut quarterly using surgical spring scissors and forceps under a binocular dissecting microscope to make a butterfly-tie shape. Paraffin sections were used for nuclear immunostaining to identify Zeb1+, Ki67+ (cell proliferation) and TUNEL+ (cell death) cells. Cryosections were used for immunostaining to quantitatively evaluate Tnfa+, Cdh1+ and Vim+ areas of the cornea. The primary antibodies and their dilutions are as followings: Zeb1 (rabbit polyclonal antiserum against Zeb1, a gift from Dr. Douglas Darling, 1:1000), anti-mouse TNF-α (BD Bioscience, Cat. # 554418, 1:100, i.e., 5 μg/ml), Ki67 (BD Pharmingen, Cat. # 550609, 1:20, i.e., 12.5 μg/ml), E-cadherin (BD Transduction Lab, Cat. # 610181, 1:50, i.e., 5 μg/ml), Vimentin (Santa Cruz Biotechnology, Cat. # sc-7557, 1:100, i.e.,

2 μg/ml) and K12 (Abclonal, Cat. # A9642, 1:50, 5 μg/ml). The respective primary antibody only and the secondary antibody only were served as negative background controls. The staining intensities of the aforementioned markers were measured using the software ImageJ and normalized to the DAPI staining in the same selected area, i.e., marker protein vs. DAPI as a percentage (%). All images were captured using the 20x lens without oil by a confocal microscope with a fixed exposure setting and the fluorescent intensities of the stained areas were quantitatively analyzed by the software ImageJ after cropping the entire cross section and setting a fixed threshold.

**Real-time quantitative PCR (qPCR)**. Total RNA was extracted using Trizol solution (Invitrogen) according to the manufacturer's instruction and the RNA content was measured by the nanodrop. The first strand cDNA was synthesized using the Invitrogen reverse transcription kit according to the manufacturer's instruction. qPCR was performed with the Strategene Mx3000P system to collect the threshold cycle (Ct) values. The double delta formula was used to calculate the expression values for each desired gene after normalized to the expression of the house keeping gene *Gapdh* or ACTB. At least 3 independent replicates were tested for statistical analysis. Primer sequences for a particular gene are selected by the online software "Primer3" at the default settings and then synthesized by the company Integrated DNA Technology (IDT) (Supplementary Table S1). All PCR products were verified by their size on 1.5% agarose gels.

**Chromatin immunoprecipitation (ChIP) assay**. ChIP assays were as previously described[14]. Briefly, 1% formaldehyde was used to crosslink the genomic DNA of the human uveal melanoma cell line C918 and mouse Rb-triple-KO (TKO) MEF cells whose ZEB1 expression levels are all high. The chromatin was sheared by sonication to an average length of 500–1000 bp. The rabbit polyclonal antiserum for Zeb1[47] was used for immunoprecipitation. Input was 1/10 of the initial amount of chromatin used to bind to the anti-Zeb1 serum. Equal amount of the pre-serum (IgG) was used as a negative control whereas the pan histone 3 antibody (abcam Cat. # ab176842) was used as a positive control. The primer sequences and their amplicon size for the ChIP-PCR are listed in Supplementary Table S2.

**Terminal deoxynucleotidyl transferase dUTP nick end labeling (TUNEL)**. Paraffin sections were prepared and the TUNEL assays were performed using the DeadEnd Fluorometric TUNEL System (Promega, Cat. # G3250) according to the manufacturer's instruction.

**Statistics and reproducibility**. Student's *t*-tests were conducted for two independent animal or cell sample comparison after an *F*-test confirmation that the comparable samples have an equal level of variance. The animal sample size for all corneal section immunostaining is $n = 3$ while the cell culture sample size is either $n = 4$ for the mouse primary corneal epithelial cell migration assay or $n = 4$ for the hTECpi cell migration assay. All values in the graphs are presented as means ± standard deviations. "***" indicates *p*-value ≤ 0.001, "**" indicates *p*-value ≤ 0.01, whereas "*" indicates *p*-value ≤ 0.05. For in vitro studies, results were obtained from at least 3 independent experiments of three technical replicates.

**Reporting summary**. Further information on research design is available in the Nature Portfolio Reporting Summary linked to this article.

## Data availability
Source data for the figures in the manuscript can be found in Supplementary data.

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

## Acknowledgements

This work was supported partly by the National Natural Science Foundation of China (82171032 L.Z.), the Natural Science Foundation of Liaoling Province (201800209, 201602210, and 20180550976 to L.Z.), National Eye Institute (R01EY026158 and R01EY030933 to D.C.D., EY026509 and EY028911 to B.P.C.), the Department of Defense (MTEC-22-02-MPAI-005 to Y.L.), National Institute of General Medical Sciences (P20GM103453 to Y.L.), James Graham Brown Cancer Center of University of Louisville Directed Gift Pilot Project Program (G1779 to Y.L.).

## Author contributions

Y.Z. and K.K.D.: performed most of the experiments, collected and analyzed the data, wrote the manuscript; F.W.: performed an additional experiment for validation; X.L. and J.Y.L.: assisted in cell culture, RNA extraction and western blot analysis; B.P.C., L.Z., and C.L.: contributed critical resources and helped analyzing the data; D.C.D. and Y.L.: designed experiments, analyzed the data, and wrote the manuscript. All authors read and approved the final manuscript.

## Competing interests

The authors declare no competing interests.
