## [Peer Review File · Communications Biology]

Reviewers' comments:

Reviewer #1 (Remarks to the Author):

Authors et al report on role of Zeb1 in corneal epithelial wound repair. Specifically, the authors assess the involvement of Zeb1 in moderating the TNF/TNFR1 pathway to induce genes beneficial for corneal epithelial cell proliferation and migration. This is an important area of research with a potential significant impact. However, there are several revisions that are recommended to strengthen the paper:

1. Use of Zeb^{-/-} wt vs Zeb^{+/+} and Zeb1 het vs Zeb^{+/-} vs monoallelic deletion – be consistent with use of terminology. Please define one in introduction and then continue to use that for consistency, as it can be confusing. Also, make clear distinction between KO vs KD when beginning to discuss ZEB1-KD.
2. Introduction – make clear early on that Zeb1^{-/+} only results in partial loss of Zeb1 expression as relevant for Fig 4.
3. Figure 1b - Quantitation of % wound area requires further detail on how the wound area (fluorescein-stained corneas) were analysed in the methods section.
4. Figure 2a, 3a, 4a – Although these current low-powered images (250um) provide a whole view of the cornea, the small high-powered images within 0d are more informative to the specific staining that has been conducted and the structures that are being spoken about eg, epithelium, endothelium and stroma.
5. Figure 4a - require structural annotation similar to Figure 5a - where is limbus? This area needs to be marked as within results it is mentioned 'Ki67+ cells detected close to limbus'.
6. Figure 2d – gene expression of TNFa in cultured primary epithelium shows no difference whilst in Figure 3a, a lot of TNFa staining is observed. However, it is challenging to say from these images where TNFa localises. Also, what timepoint was gene expression conducted in Fig 2d? Have mRNA expression been assessed across time?
7. Line 29 – 33: Again, not clear about movement of TNFa
8. A few grammatical errors across the document, please perform careful proof-read. Eg. Line 13, there is a t in implying.
9. Line 5 – 13 - For the question of whether 'cellular homeostasis 12 in the Zeb1^{-/+} epithelium is slower than that in the Zeb1^{+/+} epithelium.' - have you measured Ki67+ expression of zeb^{-/+} at steady-state without injury?
10. Line 6 – indicate that this is human cell line ZEB1-KD hTCEpi cells for clarity.
11. Methods section IHC – what is positive and negative control? Were all images, and how was the threshold for intensity of staining across the group determined? Was this conducted blinded?
12. Figure 5 a – top images are quite blurry, and difficulty to see staining and annotations.
13. How is stroma/stromal cell in cornea defined? What markers are used?

Overall, I commend the authors on the elegant series of experiments conducted for this study. Although Zeb1 itself and its functions is not novel, the specific regulation of TNF/TNFR1 pathway within corneal wound healing is new. The novelty of the findings could be highlighted to greater extent within the introduction and discussion. This is an informative body of work and will be well received by the ocular immunology and vision sciences field.

Reviewer #2 (Remarks to the Author):

The manuscript by Zhang and colleagues deals with Zeb1 involvement in corneal epithelial wound healing via its actions on cell migration and cell survival. The manuscript is well written and has ample background to inform the reader. The data is comprehensive and the design is appropriate to address the functions and mechanism of action of Zeb1 to the process of wound healing. However, all that has been presented is known from other systems and approaches that deal with corneal wound healing; the novelty here being the actions of Zeb1. The authors use a wealth of approaches to acquire the data however there appears to be several limitations in the presentation and analyses that need to be addressed, for example, the use of tissue sections over whole corneas, the lack of reagent and antibody controls for the IF, and the quantification of the IF.

I have the following major and minor issue for the authors to resolve:

1. Materials and Methods: page 17, section 1. The number of mice used in each experiment and in total from each genotype should be stated either in this section or in the figure legends. Here, it is stated that eight 3-month-old mice of each genotype were used. This might not be accurate, and the authors should double check this number.
2. Materials and Methods: page 17, section 1. The authors used a mechanical debridement model using an Alger brush and clinical imaging with dye was used as acknowledgement that the entire epithelium was removed within the demarcated zone. However, I suggest histology be done to confirm this is the case.
3. Materials and Methods: page 17, section 2. There seems to be a disconnect here. On the one hand 3month old mice were used in the in vivo study but in this section 3 week old mice were used to harvest corneas for cultivation. Please explain and discuss the rationale.
4. Materials and Methods: page 17, section 5. Here tissue sections were used to collate the data. This reviewer deems this as an inaccurate approach compared to the use of whole corneas. I suggest a subset of data be conducted with this approach in mind.
5. Materials and Methods: page 17, section 5. Please specify all Ab final concentrations as well as the dilution factor.
6. Results, page 5, Figure 2. No cellular details can be seen in this Figure, only that which is displayed in the two inset panels. I suggest that higher magnification inset be present for each main panel. There is also mention in the results commentary (page 6 , lines 5-18) and in Fig 2 (panel c) of Tunnel + cells in the stroma, but because of the low magnification images, this detail cannot be seen. Like wise on page 6 line 33 there is mention of endothelial cell death but no data is presented.
7. Results, page 7, Figure 3. Please label the wounded region or the wound margin especially in the upper panel. Please provide Ab specificity data i.e., section with out primary Ab and sections with an appropriate isotype control Ab. Please provide high power images like the insets in the lower panel or even larger this is so the reader can appreciate the extent and cell localization of the Tnfa.
8. Results, page 7, Figure 3. The Tnfa IF should be compared to the reactivity on the respective controls, not the DAPI channel. This is especially pertinent to the fact that cell death is occurring and cell nuclei might not be a valid structure for standardization.
9. Results, page 7, Figure 3. In relation to panel d, it is apparent that the Tnfa in endothelium of the control wt cornea is much higher than the het +/- especially at day 14, yet this is not indicated by the graphical representation.
10. Results, page 8, Figure Supplemental 1, lines 15-19. These cells may have an epithelial identity but were they of a corneal epithelial lineage? The authors should conduct K12 staining for each line used.
11. Results, page 9, Figure 4. Again, all IF images are difficult to decipher any details. I suggest larger panels like the inset or larger be presented for both the wt and the het +/- .
12. Results, page 10-11, section 7. The authors make statements which have not been substantiated. They need to delineate one way or the other how cells migrate in their system. Is it cell sheet movement or is it movement of individual cells or is it both. There are no visuals of how this occurs in vivo. This is a key experiment that needs to be performed.
13. Results, page 13, Figure 6. The authors need to confirm that the +/+ and the +/- cells are from the mice of the same age and are of the same passage number.

We thank the referees for their considerable interest in reviewing our work and their comments to improve our manuscript. As both reviewers' suggestion, we added two more high-powered image inserts in Figure 2a, 3a, 4a and other changes in text. The following is our point-by-point response to each of their comments and the revised text is highlighted in yellow background here and in the manuscript.

Reviewer #1 (Remarks to the Author):

Authors et al report on role of Zeb1 in corneal epithelial wound repair. Specifically, the authors assess the involvement of Zeb1 in moderating the TNF/TNFR1 pathway to induce genes beneficial for corneal epithelial cell proliferation and migration. This is an important area of research with a potential significant impact. However, there are several revisions that are recommended to strengthen the paper:

1. Use of Zeb^{-/-} wt vs Zeb^{+/+} and Zeb1 het vs Zeb^{+/-} vs monoallelic deletion – be consistent with use of terminology. Please define one in introduction and then continue to use that for consistency, as it can be confusing. Also, make clear distinction between KO vs KD when beginning to discuss ZEB1-KD.

We prefer more consistently use Zeb1^{+/+} and Zeb^{-/-} for wt and het in the text to avoid any confusing. At the same time, we have added “wt, Zeb1^{+/+}; het, Zeb1^{-/-}” in the figure legends to clarify figure labels. In addition, we have rechecked the definition of both KO and KD to make a clear distinction between them. Thank you!

2. Introduction – make clear early on that Zeb1^{+/-} only results in partial loss of Zeb1 expression as relevant for Fig 4.

We have added the sentence “The monoallelic knockout (KO) of Zeb1 (Zeb1^{+/-}) that results in partial loss of Zeb1 expression in mouse corneal epithelial cells...” in the Introduction (page 3, line 2 – 3). – Thanks!

3. Figure 1b - Quantitation of % wound area requires further detail on how the wound area (fluorescein-stained corneas) were analysed in the methods section.

We have revised the last sentence of the first section in Methods (page 17, line 17 – 19) to “The fluorescein intensity of wound area was measured using the software ImageJ after cropping the entire corneal area and setting a fixed threshold and analyzed by the software GraphPad Prism.” – Thank you!

4. Figure 2a, 3a, 4a – Although these current low-powered images (250um) provide a whole view of the cornea, the small high-powered images within 0d are more informative to the specific staining that has been conducted and the structures that are being spoken about eg, epithelium, endothelium and stroma.

We tried to include more high-power images for both wt and het corneas before the debridement (0d) as suggested; but the staining of 0d sections was weak and did not provide more information in terms of comparison between wt and het corneas. Therefore, we decide to keep the main panel of 0d in Figure 2a and Figure 3a as is, while added a high-powered

image insert for the main panel of 0d in Figure 4a as suggested. To see more detail histological structure, Figure 5a – b, Figure S2 and S3 provide some examples.

5. Figure 4a - require structural annotation similar to Figure 5a - where is limbus? This area needs to be marked as within results it is mentioned 'Ki67+ cells detected close to limbus'.
As suggested above in question 4, we have added a small high-powered image near the limbus area in Figure 4a. In Figure 2a, 3a, 4a, the limbal epithelial areas are both ends of the whole cornea sections.
6. Figure 2d – gene expression of TNFa in cultured primary epithelium shows no difference whilst in Figure 3a, a lot of TNFa staining is observed.
In Figure 2d, it was not surprising that we could not detect (nd) any Tnfa expression in the cultured mouse epithelial cells because the epithelium expressed little Tnfa in vivo before the debridement (0d) as shown in Figure 2a and the cultured corneal epithelial cells were not stimulated to express Tnfa.
However, it is challenging to say from these images where TNFa localises. Also, what timepoint was gene expression conducted in Fig 2d? Have mRNA expression been assessed across time?
We only used the cultured primary epithelial cells isolated from unwounded corneas.
7. Line 29 – 33: Again, not clear about movement of TNFa
Probably, we used a wrong word “moving”. To avoid such a misunderstanding, we delete “when moving” from the text (page 6, line 29). – Thanks!
8. A few grammatical errors across the document, please perform careful proof-read. Eg. Line 13, there is a t in implying.
We carefully proof-read more times, corrected the typo error “implyting” by “implying” (page 8, line 9). – Thanks!
9. Line 5 – 13 - For the question of whether 'cellular homeostasis 12 in the Zeb1-/+ epithelium is slower than that in the Zeb1+/+ epithelium.' - have you measured Ki67+ expression of zeb-/+ at steady-state without injury?
Yes, we measured Ki67+ expression in both wt and het corneas before the debridement (see Figure 4c).
10. Line 6 – indicate that this is human cell line ZEB1-KD hTCEpi cells for clarity.
Yes, this is the human cell line ZEB1-KD hTCEpi cells.
11. Methods section IHC – what is positive and negative control?
All the antibodies used in the study are routinely used in the lab and verified to work with mouse tissues (see the references 9, 10, 13, 14, 31, 37). The corneal sections of 0.25d after the debridement were usually served as a positive control while the primary antibody only and the secondary antibody only were served as negative background controls. To clarify this, we have added a sentence in the Methods (page 18, line 14 – 16) “The respective primary antibody only and the secondary antibody only were served as negative background controls.” – Thanks!

Were all images, and how was the threshold for intensity of staining across the group determined?

We revised the last sentence of section 5 in Methods (page 18, line 18 – 20) to “All images were captured by a confocal microscope with a fixed exposure setting and the fluorescent intensities of the stained areas were quantitatively analyzed by the software ImageJ after cropping the entire desired area and setting a fixed threshold across the group determined”. – Thanks!

Was this conducted blinded?

Yes, all the observations were conducted blinded by at least 2 persons. We revised the sentences in the Methods (page 17, line 15 – 16) to “The epithelial wound was evaluated by photography in a blinded manner...”. – Thanks!

12. Figure 5 a – top images are quite blurry, and difficulty to see staining and annotations.

We have selected the best representative images – thank you for understanding.

13. How is stroma/stromal cell in cornea defined? What markers are used?

The stroma is relatively easier to define, i.e., the middle part between the upper cellular layer epithelium and the bottom single-cell-layer endothelium. However, it is relatively difficult to differentiate the keratocytes (stromal fibroblasts) from the infiltrated immune cells. We usually define keratocytes in vivo by their evenly scattered location in the stroma and their elongated spindle nucleus shape which is different from that of the infiltrated immune cells. In addition, we did try to use the keratocyte-specific marker keratan sulfate; but it was not working.

Overall, I commend the authors on the elegant series of experiments conducted for this study. Although Zeb1 itself and its functions is not novel, the specific regulation of TNF/TNFR1 pathway within corneal wound healing is new. The novelty of the findings could be highlighted to greater extent within the introduction and discussion. This is an informative body of work and will be well received by the ocular immunology and vision sciences field.

We truly appreciate the reviewer’s comments and contribution to the improvement of the manuscript. – Thank you!

Reviewer #2 (Remarks to the Author):

The manuscript by Zhang and colleagues deals with Zeb1 involvement in corneal epithelial wound healing via its actions on cell migration and cell survival. The manuscript is well written and has ample background to inform the reader. The data is comprehensive and the design is appropriate to address the functions and mechanism of action of Zeb1 to the process of wound healing. However, all that has been presented is known from other systems and approaches that deal with corneal wound healing; the novelty here being the actions of Zeb1. The authors use a wealth of approaches to acquire the data however there appears to be several limitations in the presentation and analyses that need to be addressed, for example, the use of tissue sections over whole corneas, the lack of reagent and antibody controls for the IF, and the quantification of the IF.

I have the following major and minor issue for the authors to resolve:

1. Materials and Methods: page 17, section 1. The number of mice used in each experiment and in total from each genotype should be stated either in this section or in the figure legends. Here, it is stated that eight 3-month-old mice of each genotype were used. This might not be accurate, and the authors should double check this number.

We have rechecked it and it is correct. But it may be confusing because we used the same mice for corneal wound healing evaluation at multiple time points – thanks!

2. Materials and Methods: page 17, section 1. The authors used a mechanical debridement model using an Alger brush and clinical imaging with dye was used as acknowledgement that the entire epithelium was removed within the demarcated zone. However, I suggest histology be done to confirm this is the case.

We did not claim that the entire epithelium was removed though we tried to do so. And yes, we did check it after the debridement by histology and found a few small islands of the remained basal cells were scattered in the debrided area. - Thanks for the comments!

3. Materials and Methods: page 17, section 2. There seems to be a disconnect here. On the one hand 3 month old mice were used in the in vivo study but in this section 3 week old mice were used to harvest corneas for cultivation. Please explain and discuss the rationale.

It is extremely difficult to expand mouse primary corneal epithelial cells in culture. We found that corneal epithelial cells harvested from younger pups could passage more times. We therefore added a new sentence (page 17, line 21 – 22) “Corneal epithelial cells from younger animals could be passaged more times than older animals” in the Methods as suggested. – Thank you!

4. Materials and Methods: page 17, section 5. Here tissue sections were used to collate the data. This reviewer deems this as an inaccurate approach compared to the use of whole corneas. I suggest a subset of data be conducted with this approach in mind.

We agree with the reviewer to some extent. Using the whole cornea to assess the corneal wound healing after the debridement is a standard way in many labs. However, using the whole cornea has an obvious limitation in detail molecular analysis. This is the reason why we used corneal sections to collate the data. We appreciate your suggestion on possibly using whole corneas to collate molecular data. – Thank you!

5. Materials and Methods: page 17, section 5. Please specify all Ab final concentrations as well as the dilution factor.

We feel specifying both final concentration as well as the dilution factor for each Ab is redundant as they are easily converted to each other when the manufacturer’s catalog number is provided.

6. Results, page 5, Figure 2. No cellular details can be seen in this Figure, only that which is displayed in the two inset panels. I suggest that higher magnification inset be present for each main panel.

We did try to include more high-powered images for each main panel as suggested; but we found it interrupts the overall panel appearance since the staining for most main panels

was weak except for 0.25d and 1d time points. So, we select the main panels of 0.25d after the debridement for the high-powered image inserts to detail their histological structure. There is also mention in the results commentary (page 6 , lines 5-18) and in Fig 2 (panel c) of Tunnel + cells in the stroma, but because of the low magnification images, this detail cannot be seen. Like wise on page 6 line 33 there is mention of endothelial cell death but no data is presented.

As suggested by the reviewer, we added two more high-powered image inserts in the main panels of 0.25d after the debridement to detail both the stroma and the endothelium in Figure 4a. – Thanks!

7. Results, page 7, Figure 3. Please label the wounded region or the wound margin especially in the upper panel.

We added the labels for the wound margins (inverted white triangles) in the main panels of 0.25d after the debridement as suggested. But keep in mind that was not the original wound borders because the repair had immediately occurred and moved towards the central corneal area. 0d (the upper panel) here was before the debridement.

Please provide Ab specificity data i.e., section with out primary Ab and sections with an appropriate isotype control Ab.

All antibodies used in this study were routinely used in the lab and worked well with mouse tissues including the cornea. For background controls, we always included primary antibody only and secondary antibody only controls (i.e., we only took one picture for each control), which we will present them in the supplementary information as the journal requested. – Thank you!

Please provide high power images like the insets in the lower panel or even larger this is so the reader can appreciate the extent and cell localization of the Tnfa.

As we explained above in response to question 6, the addition of too many high-powered image inserts would interrupt the overall panel appearance since the staining for most main panels was weak except for 0.25d and 1d time points. However, to detail the localization of Tnfa, we added two more high-powered image inserts for the main panels of 1d after the debridement. – Thanks!

8. Results, page 7, Figure 3. The Tnfa IF should be compared to the reactivity on the respective controls, not the DAPI channel. This is especially pertinent to the fact that cell death is occurring and cell nuclei might not be a valid structure for standardization.

We appreciate the reviewer's comments. Using the reactivity on the respective controls as the denominator would be theoretically more accurate than DAPI for comparison; but it was impossible in practice because the background controls showed no staining under the fixed threshold setting and the number zero cannot be a denominator.

9. Results, page 7, Figure 3. In relation to panel d, it is apparent that the Tnfa in endothelium of the control wt cornea is much higher than the het +/- especially at day 14, yet this is not indicated by the graphical representation.

We acknowledge this seemingly inconsistency. But, keep in mind three corneas were actually assessed. We selected this representative het cornea as it was complete while the other two corneas were broken and incomplete. – Thank you for understanding!

10. Results, page 8, Figure Supplemental 1, lines 15-19. These cells may have an epithelial identity but were they of a corneal epithelial lineage? The authors should conduct K12 staining for each line used.

We identified the epithelial cells isolated from mouse corneas using a standard epithelial marker E-cadherin (Cdh1) and based on their morphology. We did try to stain with Krt3 and Krt4 antibodies, but none of them worked. We can try with Krt12 in future as the reviewer suggested. – Thanks!

11. Results, page 9, Figure 4. Again, all IF images are difficult to decipher any details. I suggest larger panels like the inset or larger be presented for both the wt and the het +/- .

Since the epithelial cell proliferation (Ki67+ cells) rates of both Zeb1 wt and het corneas were similar throughout the first 3 days in response to the debridement (see Figure 4c), adding main panels of het corneas does not provide more information. However, we added a high-powered image insert in the 0d panel to show a relatively quiescent epithelium situation before the debridement as the reviewer suggested. And more high-powered images of Ki67 staining are provided in Supplemental Figure S2. – Thanks!

12. Results, page 10-11, section 7. The authors make statements which have not been substantiated. They need to delineate one way or the other how cells migrate in their system. Is it cell sheet movement or is it movement of individual cells or is it both. There are no visuals of how this occurs in vivo. This is a key experiment that needs to be performed.

We agree with the reviewer's comments and we only intended to determine whether or not Zeb1 is involved in regulation of corneal epithelial cell migration after the debridement and did not intend to verify which or both ways the corneal epithelial cell migrate in vivo. So, we revised the sentence (page 11, line 9 – 11) to "To determine how E-cadherin and Vimentin, the well-known cell-to-cell connection components regulated by Zeb1, are changed in expression over the recovery of the denuded epithelium..." – Thank you!

13. Results, page 13, Figure 6. The authors need to confirm that the +/+ and the +/- cells are from the mice of the same age and are of the same passage number.

We have indicated in the Methods that mouse corneal epithelial cells were isolated from 3 – 4 weeks old pups of both Zeb1+/+ and Zeb1+/- genotypes and the hTCEpi cells of the vector control and ZEB1sh were from the same passage. – Thanks!

Reviewers' comments:

Reviewer #1 (Remarks to the Author):

In response to comment 4:

- Figure 2A – the organisation of the low and high-powered images could be improved – the high-powered magnification obscures the first panel (0d). I understand that this was done as there was limited space on the figure it is derived from. But it would be better to have the high-powered image next to the respective panels that the magnified image is taken from (0.25d) for clarity.
- In addition, the magnification (x40 etc) of the two images should be noted in the figure legend and for all other figures.
- Figure 3A is fine as is because the high-powered images are on the respective panels they correspond to.

In response to comment 13:

- That is fine, however in the introduction it would be worth defining what the corneal stroma comprises of for greater clarity.

Reviewer #2 (Remarks to the Author):

The manuscript by Zhang and colleagues is much improved and the authors were able to deal with most of my concerns. There are still a few issues that remain unresolved.

1. The authors provided an answer and clarification, but they did not make it clear in the manuscript. Please modify the text in the manuscript accordingly.
2. Again, the authors provide an answer to my original question but they should illuminate the reader that histology was performed and the results that came of it.
3. The authors answered satisfactorily and modified the text accordingly.
4. The authors have not answered this question to my satisfaction. Although there are molecular analyses in the manuscript, there are also clinical and biochemical analyses. As originally recommended, some analyses for validation purposed should be done on whole corneas. This will provide an accurate overview of localization and measure of one or two specific parameters.
5. Final antibody concentration is better and more accurate that stating the dilution factor; please include.
6. The authors answered satisfactorily and modified the figures accordingly.
7. The authors answered satisfactorily and modified the text accordingly or provided the rationale for no modification.
8. The authors answered satisfactorily and provided the reason for no modification.
9. The authors answered satisfactorily but I suggest that some commentary be included in the manuscript to make the reader aware of the variations between samples/mice.
10. I am not satisfied with the response to my original concern. K12 staining should be done as a marker of corneal epithelial identity. Mice do not have a functional K3 gene this is why K3 staining did not work.
11. The authors answered satisfactorily and modified the figures accordingly.
12. The authors answered satisfactorily and modified the text accordingly.
13. The authors answered satisfactorily.

Reviewer #1 (Remarks to the Author):

In response to comment 4: - Figure 2A – the organisation of the low and high-powered images could be improved – the high-powered magnification obscures the first panel (0d). I understand that this was done as there was limited space on the figure it is derived from. But it would be better to have the high-powered image next to the respective panels that the magnified image is taken from (0.25d) for clarity.

Thank you for the suggestion. We have rearranged the high-powered images next to the respective panels that the magnified image is taken from (0.25d).

- In addition, the magnification (x40 etc) of the two images should be noted in the figure legend and for all other figures.

All images were captured using the 20x lens without oil by a Nikon confocal microscope. And we have made an according modification in the Methods. Thanks!

- Figure 3A is fine as is because the high-powered images are on the respective panels they correspond to.

Thank you!

In response to comment 13: - That is fine, however in the introduction it would be worth defining what the corneal stroma comprises of for greater clarity.

Thank you for the comments. We have modified the sentence at line 27 in page 2 to “to protect the underlying stroma that mainly contains quiescent keratocytes and it may lose its transparency if damaged due to the infiltration of immune cells and the activation of the keratocytes.” – Thanks!

Reviewer #2 (Remarks to the Author):

The manuscript by Zhang and colleagues is much improved and the authors were able to deal with most of my concerns. There are still a few issues that remain unresolved.

1. The authors provided an answer and clarification, but they did not make it clear in the manuscript. Please modify the text in the manuscript accordingly.

Per request, we have revised the sentence “The epithelial wound was evaluated by photography...” to “The epithelial wound of the same mice was evaluated by photography...” in the Materials and Methods section 1. – Thanks!

2. Again, the authors provide an answer to my original question but they should illuminate the reader that histology was performed and the results that came of it.

Per request, we have included the representative histological H&E-stained sections of the corneas collected in 6 hours after the debridement in Supplemental Fig. S7 and added the sentence “The mechanical debridement was confirmed by the H&E histology at 6 h (Supplemental Fig. S7) after the debridement” in the Materials and Methods section 1. – Thanks

Supplemental Figure S7. H&E histological assessment on the debrided corneas. To clarify whether the central corneal epithelium is smoothly removed by the Alger Brush II under a stereo microscope, representative corneas of both *Zeb1*^{+/+} and *Zeb1*^{+/-} mice were collected in 6 hours (0.25d) after the mechanical debridement, and then fixed, paraffin-embedded, sectioned and H&E stained. It appeared the mechanical debridement of the corneal epithelium was done well with few basal cells scattered in the debrided areas.

3. The authors answered satisfactorily and modified the text accordingly.

Thank you!

4. The authors have not answered this question to my satisfaction. Although there are molecular analyses in the manuscript, there are also clinical and biochemical analyses. As originally recommended, some analyses for validation purposed should be done on whole corneas. This will provide an accurate overview of localization and measure of one or two specific parameters.

Per suggestion to validate the results from the corneal sections by a whole cornea staining we selectively conducted the wholemout immunostaining of both *Zeb1*^{+/+} and *Zeb1*^{+/-} corneas on day 0 (before debridement) and day 1 after the debridement to detect changes in corneal TUNEL⁺ cell death and Tnfa⁺ secretion. As a result, we obtained a similar data set (Supplemental Fig. S1) to those from the corneal sections (Fig. 2 and 3). Therefore, we added the sentence “To validate this observation, we selectively conducted a wholemout immunostaining of both *Zeb1*^{+/+} and *Zeb1*^{+/-} corneas on day 0 (before debridement) and day 1 after the debridement to detect changes in corneal TUNEL⁺ cell death and had a result (Supplemental Fig. S1a – b) similar to the corneal sections (Fig. 2a – c)” in the Results section 2 and “To confirm the result, we also immunostained

wholemount corneas with the Tnfa antibody and had a similar result (Supplemental Fig. S1a – b) to the corneal sections (Fig. 3a – c)” in the Results section 3. However, due to the restricted number of Zeb1 mutant mice, we only selected two genotypes, two time points and two most important parameters, i.e., TUNEL detection for cell death and Tnfa immunostaining for the possible causal cytokine without replicate (n=1).

Supplemental Figure S1. Wholemount immunostaining evaluation on corneal apoptotic cell death and Tnfa secretion before and after the debridement. (a) To validate the results obtained by immunohistochemistry (IHC) on the corneal cross sections (Fig. 2 and Fig. 3) we conducted a corneal wholemount immunostaining to detect TUNEL⁺ cell death and Tnfa⁺ secretion using both Zeb1^{+/+} (wt) and Zeb1^{+/-} (het) corneas collected before (0d) and 1 day (1d) after the mechanical debridement. (b) The ratios of either TUNEL⁺ or Tnfa⁺ area to the related total

corneal area indicated that the mechanical debridement and the monoallelic deletion of Zeb1 mutually increased corneal apoptotic cell death and Tnfa secretion.

5. Final antibody concentration is better and more accurate than stating the dilution factor; please include.

Per suggestion, we have added the actual concentration for each antibody used in the study. Thanks!

6. The authors answered satisfactorily and modified the figures accordingly.

Thanks!

7. The authors answered satisfactorily and modified the text accordingly or provided the rationale for no modification.

Thanks!

8. The authors answered satisfactorily and provided the reason for no modification.

Thanks!

9. The authors answered satisfactorily but I suggest that some commentary be included in the manuscript to make the reader aware of the variations between samples/mice.

Per suggestion, we have added a note in the legend of Figure 3 “The 14-day Zeb1^{+/-} cornea may not be representative as it was the only complete cornea while the other two corneas were broken and incomplete.” – Thanks!

10. I am not satisfied with the response to my original concern. K12 staining should be done as a marker of corneal epithelial identity. Mice do not have a functional K3 gene this is why K3 staining did not work.

Per suggestion, we bought a new K12 antibody from Abclonal (Cat. #: A9642) and conducted an immunofluorescence (IF) staining on the isolated mouse corneal epithelial cells (P3). It worked well as the reviewer suggested and the primary antibody only and secondary antibody only controls showed no signal above background. We included the K12-stained mouse corneal epithelial cell image in Supplemental Fig. 2S. – Thanks!

Supplemental Figure S2. Mouse corneal epithelial cell isolation and culture. The isolated mouse cornea is placed upside down on cell culture plate coated with 0.1% gelatin for **(a)** 0, **(b)** 1, **(c)** 3, and **(d)** 7 days. **(e)** The monolayer-cultured corneal epithelial cells and their immunofluorescence with **(f)** the corneal epithelial-specific marker Krt12 and **(g)** the common epithelial marker E-cadherin (Cdh1).

11. The authors answered satisfactorily and modified the figures accordingly.

Thanks!

12. The authors answered satisfactorily and modified the text accordingly.

Thanks!

13. The authors answered satisfactorily.

Thanks!

REVIEWERS' COMMENTS:

Reviewer #2 (Remarks to the Author):

The authors should be congratulated on a fine job in addressing all my concerns.